# Exploring the Relationship Between Perioperative Inflammatory Biomarkers and Oncological Recurrence in Patients Undergoing Pulmonary Cancer Surgery

**DOI:** 10.3390/cancers17071159

**Published:** 2025-03-30

**Authors:** Elena de la Fuente, Oscar Morgado, Francisco de la Gala, Elena Vara, Pilar Zuluaga, Almudena Reyes, Carlos M. Simón, Javier Hortal, Patricia Piñeiro, Ignacio Garutti

**Affiliations:** 1Anesthesiology Department, Hospital General Universitario Gregorio Marañón, 28007 Madrid, Spain; efuente@salud.madrid.org (E.d.l.F.);; 2Faculty of Medicine, Complutense University of Madrid, 28040 Madrid, Spain; 3Fundación para la Investigación del Hospital Gregorio Marañón, 28009 Madrid, Spain; 4Biochemical Department, Facultad Medicina, Universidad Complutense de Madrid, 28040 Madrid, Spain; 5Statistical Department and Operational Research, Facultad Medicina, Universidad Complutense de Madrid, 28040 Madrid, Spain; 6Thoracic Surgery Department, Hospital General Universitario Gregorio Marañon de Madrid, 28007 Madrid, Spain; 7Pharmacology Department, Facultad Medicina, Universidad Complutense de Madrid, 28040 Madrid, Spain

**Keywords:** lung cancer, recurrence, inflammation, cytokines, surgery, predictive model

## Abstract

One of the major challenges in oncologic lung surgery is the risk of disease recurrence. While several risk factors for cancer relapse are well known, the role of inflammation in oncologic recurrence has been less extensively studied. In this study, we developed two logistic regression models to predict recurrence: one based solely on clinical parameters, and another incorporating perioperative inflammatory biomarkers measured in blood and lung fluid. Our findings indicate that the inclusion of cytokine levels significantly improves the predictive accuracy of recurrence models. Therefore, we recommend measuring these biomarkers to facilitate a more tailored follow-up and treatment approach for patients undergoing surgery for primary lung cancer.

## 1. Introduction

Lung cancer represents a significant challenge in contemporary medicine, as it is a solid tumor with the highest incidence and mortality rates in both sexes worldwide [1]. Although numerous therapeutic options are available for its treatment, curative-intent surgery remains the most effective approach for early-stage disease [2,3]. However, despite achieving complete surgical resection, residual tumor cells may persist at the surgical site or disseminate undetected from the primary tumor, potentially leading to local recurrence and/or distant metastases [4]. As a result, recurrence rates remain alarmingly high and constitute one of the most critical prognostic factors in lung cancer [5].

During surgery, the body undergoes a stress response that triggers catecholamine release and activates physiological damage-repair mechanisms. This, in conjunction with perioperative immunosuppression and its subsequent inflammatory response, may promote metastatic progression even after an apparently complete surgical resection [4].

Lung cancer is closely linked to inflammation, not only because it exacerbates the pro-inflammatory state both locally and systemically, but also because pre-existing inflammatory conditions have been observed to significantly increase the risk of developing cancer or pulmonary metastases [6].

Oncologic cells that evade immune surveillance can proliferate in distant organs, eventually leading to the formation of new blood vessels to sustain their growth, the production of specific signaling molecules, and the establishment of a specialized extracellular matrix. This process gives rise to what is known as the tumor microenvironment [7]. Several studies have confirmed that the balance between pro-inflammatory and anti-inflammatory cytokines within the tumor microenvironment plays a crucial role in determining tumor behavior [6,7,8,9,10]. Tumor cells release cytokines and chemokines that attract various immune cells, which, in turn, further stimulate cytokine production, generating a positive feedback loop that amplifies the immune response. Many cytokines play a critical role in carcinogenesis [11,12]. Recent research has highlighted the pivotal roles of IL-6 and IL-8 in the development and progression of lung cancer [13]. Specifically, IL-6 has been associated with tumor proliferation in non-small-cell lung cancer (NSCLC) [13], while IL-8 has been identified as a key growth factor in lung cancer [14]. Other interleukins (e.g., IL-1, IL-10, and IL-12) and metalloproteinases have also been implicated in pulmonary oncogenesis [15,16,17,18,19,20].

Based on this evidence, we hypothesize that perioperative monitoring of inflammatory response may be associated with lung cancer recurrence in patients undergoing surgical resection. The primary objective of this study is to investigate the relationship between lung cancer recurrence and perioperative inflammatory status, assessed in both blood and bronchoalveolar lavage fluid.

## 2. Materials and Methods

A single-center retrospective cohort study was conducted using a prospective database at a tertiary hospital in Madrid, Spain. The study received approval from the Clinical Research Ethics Committee of Hospital Universitario Gregorio Marañón, and informed consent was obtained from all participants. Patients were eligible for inclusion if they had previously participated in the clinical trials NCT02168751 or EudraCT 2011-002294-29.

For this study, rigorous inclusion and exclusion criteria were established to ensure sample homogeneity and the validity of the results. Inclusion criteria encompassed legally competent patients of both sexes, aged 18 years or older, who voluntarily consented to participate by signing an informed consent form. Additionally, patients were required to be scheduled for specific surgical procedures and to have adequate pulmonary function, defined as a forced expiratory volume in the first second (FEV1) greater than 60% or a forced vital capacity (FVC) exceeding 50%.

Exclusion criteria included pregnancy or lactation, blood transfusion within the last 10 days, and contraindications to the application of protective pulmonary ventilation. Patients with heart failure classified as functional class III or IV according to the New York Heart Association criteria in the week prior to surgery, as well as those undergoing surgery for non-oncological reasons, were also excluded. A total of 146 patients were analyzed, with a mean follow-up of 10 years; among these, only 93 patients with primary lung cancer were included.

All patients received standardized perioperative care, except for the hypnotic agent used for anesthesia maintenance. In one group, anesthesia was maintained with propofol (2–3 mg·kg^−1^), while in the other, sevoflurane was used, with the goal of achieving a Bispectral Index (BIS) of 45–60 in both cases. A double-lumen endotracheal tube was inserted, and its correct placement was verified using a fiberoptic bronchoscope.

Initially, pulmonary ventilation was set with a tidal volume of 8 mL per kilogram of ideal body weight, a positive end-expiratory pressure (PEEP) of 5 cmH_2_O, an inspired oxygen fraction (FiO_2_) between 0.4 and 0.5, and a respiratory rate adjusted to maintain end-tidal carbon dioxide (EtCO_2_) levels between 30 and 35 mmHg. Intraoperative and postoperative analgesia was administered via thoracic paravertebral blocks. The clinical variables collected are presented in Table 1.

Bronchoalveolar lavage (BAL) fluid samples were obtained exclusively from the affected lung both before the start of surgery and immediately after the procedure. BAL samples were collected by advancing a fiberoptic bronchoscope into the deepest bronchi of the affected lung until encountering additional resistance. An initial 25 mL aliquot of 0.9% saline solution was instilled, with the residual fluid subsequently discarded. The procedure was then repeated with a second saline instillation, and the aspirated fluid was collected for analysis. The biomarkers analyzed included IL-1, IL-2, IL-4, IL-6, IL-7, IL-8, IL-10, IL-12, TNF-α, MCP-1, MMP-2, MMP-3, MMP-7, MMP-9, nitric oxide (NO), and carbon monoxide (CO). The balance between pro-inflammatory and anti-inflammatory markers was assessed using the IL-8/IL-10 and IL-6/IL-10 ratios [21].

Cytokine blood levels were measured from arterial blood samples collected during routine arterial blood gas analyses, as well as at 6 and 18 h post-surgery. The samples were filtered using a sterile gauze and centrifuged at 400× *g* for 15 min at 4 °C. The supernatant was stored at −20 °C until analysis in a specialized laboratory. Cytokine concentrations were determined using the ELISA technique. The biomarkers measured included IL-1, IL-2, IL-4, IL-6, IL-7, IL-8, IL-10, IL-12, TNF-α, MCP-1, MMP-2, MMP-3, MMP-7, MMP-9, NO, CO, and VEGF.

### Statistical Analysis

The results of continuous variables that did not follow a normal distribution are presented as medians with their corresponding interquartile range (IQR), while continuous variables that followed a normal distribution are expressed as means with standard deviations. Qualitative variables are reported as frequencies and percentages. Normality was assessed using the Kolmogorov–Smirnov test.

Continuous variables were compared between the recurrence and non-recurrence groups using the *t*-test for normally distributed variables and the Mann–Whitney U test for non-normally distributed variables. For qualitative variables, the Chi-square test or Fisher’s exact test was applied as appropriate.

A univariate logistic regression analysis was performed on variables obtained before, during, and after surgery (at 6 and 18 h) to predict recurrence versus non-recurrence. Following Hosmer–Lemeshow’s recommendations [22], variables with a *p*-value < 0.25 in the univariate models were considered candidates for inclusion in the multivariate models. Significant variables for the multiple logistic regression models were selected using a stepwise approach. This method systematically evaluates the inclusion or exclusion of each variable based on its contribution to model fit, using statistical criteria such as the Wald *p*-value.

Two multiple logistic regression models were developed to predict recurrence versus non-recurrence (Model 1 and Model 2). Model 1 included only significant clinical variables, while Model 2 incorporated both the clinical variables from Model 1 and significant variables identified in the univariate logistic regression analysis of blood and bronchoalveolar lavage samples.

Receiver Operating Characteristic (ROC) curves were used to assess the models’ ability to predict recurrence. The Nagelkerke R^2^ coefficient was calculated to evaluate the models’ goodness of fit and predictive power. ROC curve comparisons were conducted using the MedCalc 22.023 statistical package [23].

All statistical tests were two-tailed, and a *p*-value < 0.05 was considered statistically significant. Analyses were performed using IBM SPSS Statistics for Windows, Version 27.0 [24].

## 3. Results

### 3.1. Patients Characteristics

Of the 93 patients initially included in the study, 39 (41.9%) experienced tumor recurrence during the 10-year follow-up period. When comparing patients with and without recurrence, statistically significant differences were observed in the patient’s vital status (*p* < 0.01), tumor stage (T) (*p* = 0.022), and type of surgery performed (*p* = 0.02). The remaining preoperative clinical variables were comparable between both groups (Table 1).

### 3.2. Blood Biomarkers

Baseline cytokine levels were compared between the recurrence and non-recurrence groups, as shown in Table 2 and Figure 1. Following surgery, all pro-inflammatory cytokine levels increased at 6 h, followed by a decline at 18 h. In the recurrence group, pro-inflammatory cytokine expression at both 6 and 18 h was higher compared to the non-recurrence group. Some cytokines exhibited significantly elevated levels in patients with recurrence. In contrast, IL-4 and IL-10 levels showed no significant differences at any timepoint or between the groups. Additionally, a greater increase in metalloproteinases was observed in the non-recurrence group.

### 3.3. Bronchoalveolar Lavage Biomarkers

In the samples obtained at the beginning of surgery, before initiating one-lung ventilation (Table 3 and Figure 2), cytokine expression was similar between the recurrence and non-recurrence groups. However, after surgery, cytokine levels were generally higher in patients who experienced recurrence, except for IL-1 and MMP-9, which were elevated in the non-recurrence group. In contrast, IL-12 levels remained unchanged in both groups.

### 3.4. Predictive Factors for Recurrence

Two multivariate logistic regression models were performed. In the first model (Model 1), only clinical variables were included, and after applying stepwise selection, only tumor stage and type of surgery were retained (Table 4). Subsequently, a second model was adjusted, which, in addition to the clinical variables from Model 1, included candidate variables derived from blood and bronchoalveolar lavage. Using stepwise selection, Model 2 was developed, which retained only the type of surgery variable and incorporated the plasma variables IL-6 at 6 h, MCP-1 at 18 h, the IL-6/IL-10 ratio, basal MMP-3, and the bronchoalveolar lavage variables TNFα post-surgery, as well as basal and post-surgery MMP-9 (Table 4).

The multivariate model (Model 1) had a Nagelkerke R^2^ coefficient of 0.19, indicating a relatively low explained variability. With a cutoff point of 0.5, the sensitivity was 35.9% and the specificity was 92.3%. The area under the ROC curve (AUC) was 0.65 (95% CI: 0.53–0.77).

In contrast, the multivariate model (Model 2) showed a Nagelkerke R^2^ coefficient of 0.56. Using the same cutoff point of 0.5, sensitivity increased to 80%, while specificity was 89.6%. The AUC for Model 2 was 0.89 (95% CI: 0.81–0.97) (Figure 3).

Finally, a comparison of the ROC curves revealed a statistically significant difference between the two models (*p* < 0.0001).

## 4. Discussion

The comprehensive analysis of the collected data offers a detailed perspective on recurrence in lung cancer. Our findings indicate that measuring cytokines, in conjunction with traditional clinical variables, enhances the prediction of tumor recurrence following surgical resection. This underscores the relationship between inflammation and carcinogenesis post-surgery.

Developing accurate probability models to predict early recurrence after surgery is vital in understanding the natural history of lung cancer. Such models enable the implementation of preventive strategies immediately post-surgery, potentially delaying or mitigating disease progression and minimizing its impact on patient recovery. Additionally, they facilitate more efficient resource planning for subsequent care.

Our analyses, consistent with other published studies [2,3,5], reveal that clinical variables like tumor size and the type of surgical procedure are significant predictors of tumor recurrence. Notably, our results suggest that more aggressive surgical approaches correlate with a higher likelihood of recurrence. Conversely, other clinical variables were not statistically significant in the predictive model based solely on clinical characteristics, which contrasts with previous reports [3,25].

Regarding anesthetic protocols, our findings diverge from the study by Watanabe (2022) [26], indicating no statistically significant differences between inhaled and intravenous anesthetics in maintaining hypnosis during surgery.

Tumor resection induces a physiological stress response that triggers a pro-inflammatory state, potentially contributing to tumor recurrence [6,27]. The lungs are particularly susceptible to excessive inflammatory reactions [28]. Cytokines such as IL-1, IL-6, IL-8, and TNFα serve as inflammation mediators, rapidly increasing in response to surgical trauma, sometimes excessively [6,7,13]. Elevated cytokine levels have been linked to tumor recurrence [3,6].

Analyzing cytokines in bronchoalveolar lavage fluid is crucial for assessing the impact of surgical trauma and mechanical ventilation in lung cancer patients. Our findings, supported by previous studies [4,28], demonstrate a significant increase in cytokines in this fluid when the lung is under stress. As suggested by De la Gala et al. [20], we hypothesize that surgical trauma may lead to ischemia–reperfusion injuries, correlating with increased IL-8 and infiltration of macrophages and neutrophils [7,29].

Our study further indicates that TNFα levels measured from the operated lung at surgery’s conclusion serve as a strong predictive marker for recurrence, associated with poor prognosis, as previously noted by S.V. Baudoin [29].

Concerning cytokines in the blood of our patients, our results align with prior research indicating that elevated postoperative IL-6 levels correlate with higher recurrence rates [13]. We believe that monitoring IL-6 in blood could be valuable for predicting recurrence. Similarly, as noted by K. Kwasniak [17], stable levels of IL-4 and IL-10 in both patient groups suggest a potential protective role for these cytokines.

Additionally, other biomarkers such as metalloproteinases, specifically those derived from bronchoalveolar lavage samples, warrant attention. The literature shows that elevated baseline levels of MMP2 are associated with lung damage, while increased MMP9 levels may offer protective benefits against lung injury [19,28]. Our analysis supports these findings, suggesting that baseline measurements of these metalloproteinases could provide valuable prognostic information.

By evaluating the balance between pro-inflammatory and anti-inflammatory states in bronchoalveolar fluids, we conclude that assessing these balances may be highly relevant for predicting tumor recurrence. In line with J. Sun’s findings [21], we believe that a balanced state could indicate a favorable postoperative prognosis; our data suggest that an increase in the pro-inflammatory state (IL-6/IL-10 ratio) is a strong predictor of tumor recurrence.

The evaluation of these biomarkers enhances our predictive capacity for determining tumor recurrence probability in lung cancer patients. Considering the timeframe before the first clinical manifestations arise, we could implement early interventions, such as chemotherapy to prevent tumor cell dissemination, immunotherapy to mitigate inflammatory effects, or rigorous follow-up for high-risk patients.

Recurrences in cancer impose a significant burden on healthcare systems; therefore, early detection strategies can facilitate more efficient resource use.

Our study does have limitations. It is a single-center study conducted at Gregorio Marañón University General Hospital. Although derived from a randomized controlled trial, our predictive models were retrospective. When utilizing predictive models based on cytokines, it is essential to consider the temporal variations in cytokine expression, as well as to employ biomarkers that respond rapidly to surgical insults. A notable limitation is that cytokine levels in bronchoalveolar fluid were only measured during surgery; analyzing this fluid postoperatively could have yielded additional insights. However, this approach is typically not recommended in this clinical scenario. Furthermore, we cannot ascertain whether the results would differ with the inclusion of patients with underlying lung disease or non-primary lung tumors, or if modifications to mechanical ventilation conditions were applied.

## 5. Conclusions

In conclusion, there is a clear association between perioperative inflammatory markers—both blood-based and those obtained from bronchoalveolar lavage fluid—and tumor recurrence following pulmonary oncological surgery. Moreover, in conjunction with established clinical risk factors for oncological recurrence, inflammatory biomarkers significantly enhance the prediction of oncological relapse. The integration of biological and clinical models facilitates the development of more specific and sensitive predictive frameworks.

## Figures and Tables

**Figure 1 cancers-17-01159-f001:**
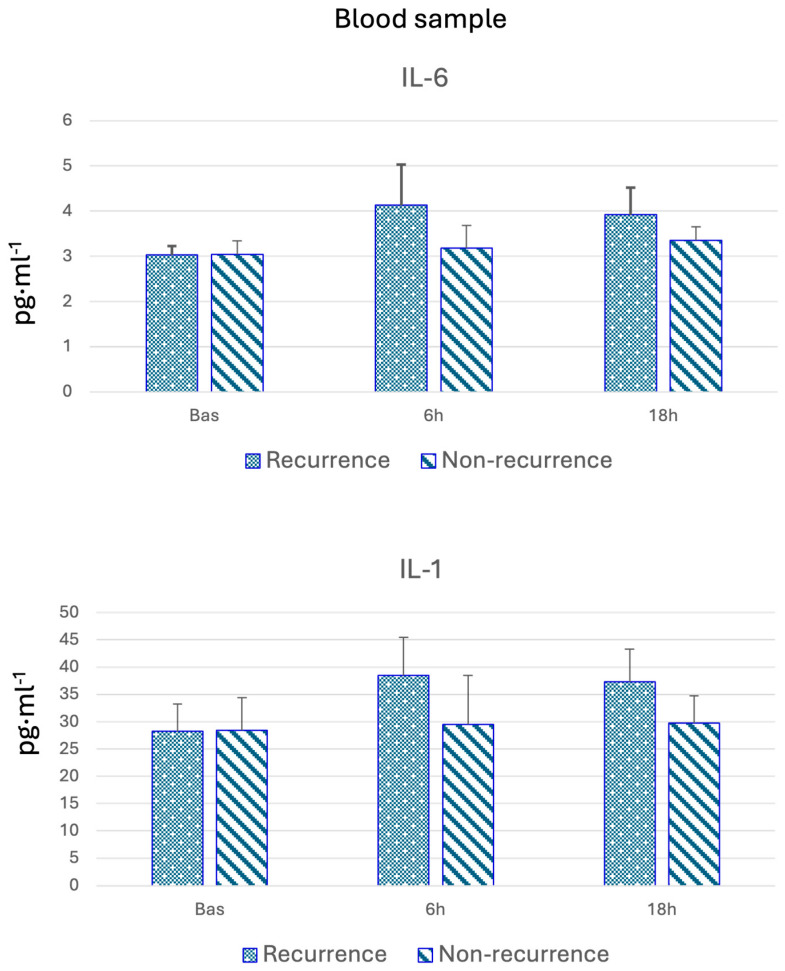
Blood samples of IL-6 and IL-1.

**Figure 2 cancers-17-01159-f002:**
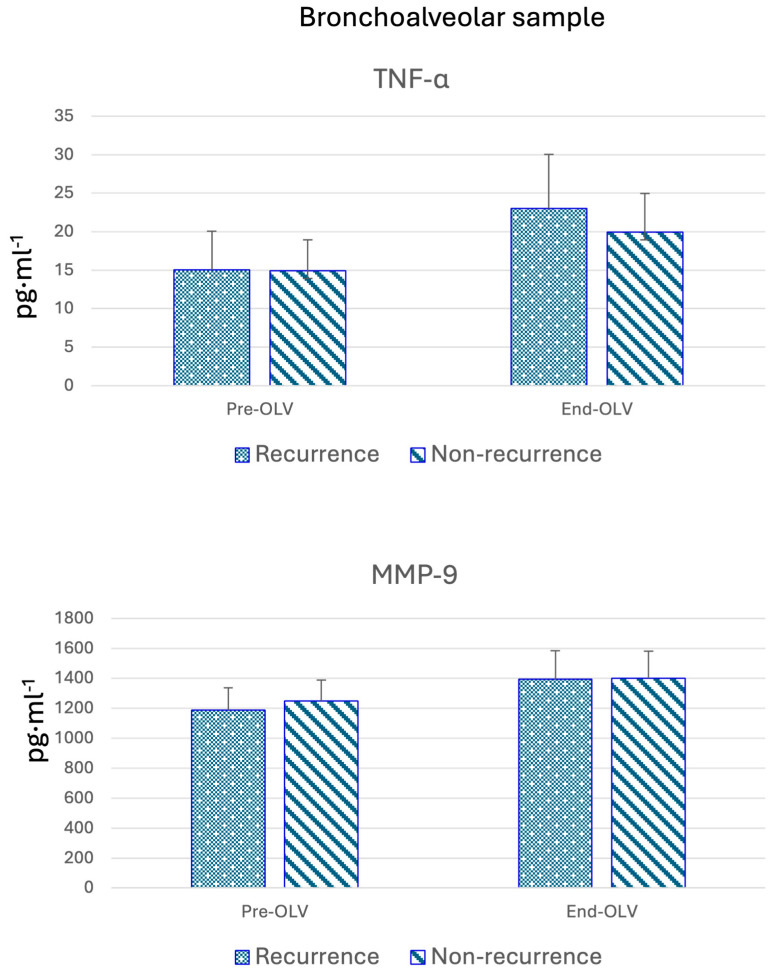
Median bronchoalveolar lavage values of TNF-α and MMP-9. Pre-OLV and End-OLV: Before and after one-lung ventilation.

**Figure 3 cancers-17-01159-f003:**
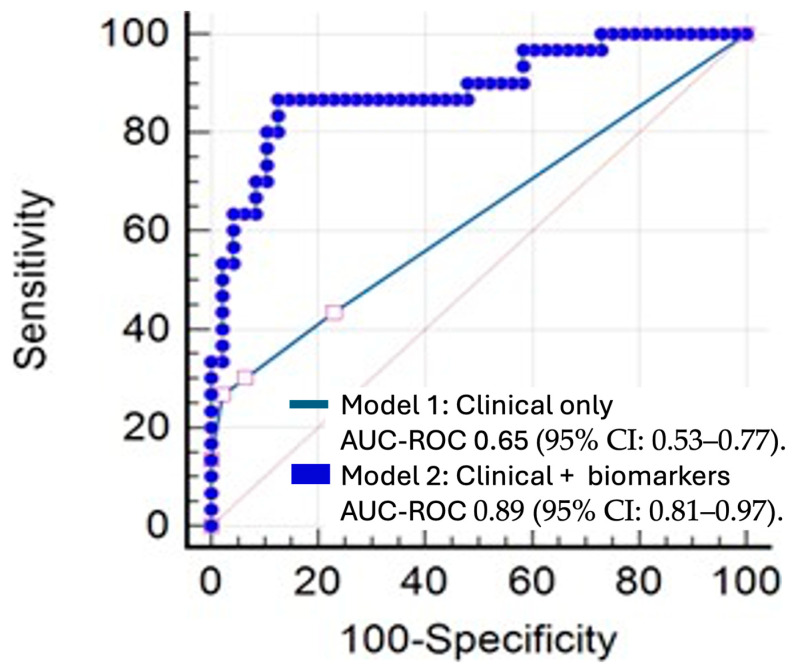
Comparison of ROC curves for logistic regression models 1 and 2. Model 1: clinical variables only. Model 2: clinical variables and inflammatory markers.

**Table 1 cancers-17-01159-t001:** Clinical and demographic variables.

Variable	No Recurrence (*n* = 54)	Recurrence (*n* = 39)	*p*-Value
Anesthetic Group			0.336
Propofol	25	22	
Inhalational	29	17	
Status			<0.01
Deceased	18	37	
Alive	36	2	
Type of Associated Complications			0.364
No complications	27	15	
Minor complications	21	16	
Major complications	6	8	
Staging (T)			0.022
T1–T2	52	32	
T3–T4	2	7	
Medical Complications			0.238
Yes	20	20	
No	30	18	
Surgical Complications			0.754
Yes	15	12	
No	39	27	
PPC			0.653
Yes	13	11	
No	41	28	
Sex			0.13
Female	22	10	
Male	32	29	
Tobacco Use			0.252
Never	20	11	
Former smoker *>* 6 months	21	11	
Former smoker < 6 months	7	8	
Smoker	6	9	
Anesthetic Risk			0.328
ASA I	2	2	
ASA II	32	17	
ASA III	20	20	
VATS			0.659
Open	4	2	
Closed	50	37	
Surgery			0.021
Pneumonectomy/Bilobectomy	2	8	
Lobectomy	42	22	
Segmentectomy	10	9	
Age (years)	65.5 (60–73.75)	65 (59–72.5)	0.427
Weight (kg)	70 (64.5–80.75)	70 (62–80.5)	0.604
Height (cm)	166 (160–172)	167 (160–173)	0.679
BMI	26 (23.9–28.6)	24.57 (22.69–28.4)	0.167

ASA: American Society of Anesthesia; VATS: video-assisted thoracoscopy; BMI: body mass index; PPC: postoperative pulmonary complications.

**Table 2 cancers-17-01159-t002:** Cytokine blood level obtained before and after surgery.

Variable	RECURRENCE	Basal	*p*-Value	Postoperative 6 h	*p*-Value	Postoperative 18 h	*p*-Value
		Median (IQR)		Median (IQR)		Median (IQR)	
TNFα (pg·mL^−1^)	Yes	7.11 (6.38–7.36)	0.90	9.28 (8.02–11.54)	0.14	8.68 (7.60–9.59)	0.18
	No	7.01 (6.59–7.53)		8.44 (7.58–10.92)		8.22 (7.45–9.45)	
IL-1 (pg·mL^−1^)	Yes	28.23 (24.55–31.10)	0.45	38.48 (24.73–44.13)	0.55	37.30 (25.17–42.35)	0.57
	No	28.44 (24.59–32.35)		29.51 (24.22–45.47)		29.74 (26.18–42.89)	
IL-2 (pg·mL^−1^)	Yes	0.86 (0.81–0.88)	0.93	1.17 (0.97–1.43)	0.14	1.24 (0.94–1.43)	0.34
	No	0.85 (0.80–0.90)		0.98 (0.92–1.39)		0.98 (0.93–1.38)	
IL-4 (pg·mL^−1^)	Yes	0.33 (0.30–0.38)	0.57	0.37 (0.35–0.40)	0.76	0.39 (0.37–0.47)	0.39
	No	0.33 (0.30–0.38)		0.37 (0.35–0.40)		0.39 (0.37–0.47)	
IL-6 (pg·mL^−1^)	Yes	3.03 (2.83–3.14)	0.41	4.13 (3.06–5.09)	0.03	3.92 (3.09–4.43)	0.23
	No	3.04 (2.90–3.20)		3.18 (2.86–4.89)		3.35 (2.90–4.35)	
IL-7 (pg·mL^−1^)	Yes	2.80 (2.49–3.12)	0.42	6.03 (4.01–7.84)	0.09	4.26 (3.47–4.65)	0.42
	No	2.75 (2.40–3.01)		4.21 (3.90–7.03)		3.88 (3.05–4.66)	
IL-8 (pg·mL^−1^)	Yes	0.93 (0.63–1.23)	0.74	16.01 (7.20–23.93)	0.60	2.48 (1.48–3.33)	0.31
	No	0.97 (0.70–1.08)		9.85 (7.24–22.79)		2.07 (1.15–3.19)	
IL-10 (pg·mL^−1^)	Yes	0.09 (0.08–0.10)	0.53	0.09 (0.08–0.10)	0.82	0.09 (0.08–0.10)	0.73
	No	0.09 (0.08–0.10)		0.09 (0.08–0.10)		0.09 (0.08–0.10)	
MCP1 (pg·mL^−1^)	Yes	248.62 (211.12–268.19)	0.19	373.14 (347.59–395.09)	0.16	377.31 (356.61–396.45)	0.18
	No	234.04 (202.54–253.31)		356.15 (328.99–381.76)		364.62 (322.93–394.79)	
NO (mmHg)	Yes	31.29 (29.35–33.34)	0.33	27.48 (25.42–30.96)	0.64	28.45 (25.98–31.21)	0.70
	No	30.82 (28.41–33.13)		28.25 (25.86–30.63)		29.10 (25.70–30.86)	
CO (mmHg)	Yes	2.61 (2.37–2.88)	0.41	2.77 (2.69–3.04)	0.59	2.86 (2.75–2.98)	0.20
	No	2.66 (2.48–2.86)		2.81 (2.70–3.02)		2.92 (2.81–3.03)	
IL-6/IL-10	Yes	34.12 (30.70–42.04)	0.39	48.34 (32.55–55.24)	0.10	42.63 (34.98–46.98)	0.26
	No	34.58 (30.44–37.18)		41.08 (29.44–50.69)		40.36 (30.54–47.45)	
IL-8/IL-10	Yes	10.25 (6.60–14.57)	1.00	118.56 (76.68–231.73)	0.48	24.90 (11.97–36.62)	0.34
	No	10.67 (8.15–12.49)		166.61 (83.65–253.82)		26.51 (16.44–35.38)	
MMP2 (pg·mL^−1^)	Yes	2.22 (1.94–2.48)	0.71	4.06 (3.77–6.19)	0.93	5.00 (4.42–6.21)	0.72
	No	2.21 (1.96–2.48)		4.99 (3.69–6.24)		5.44 (4.33–6.20)	
MMP3 (pg·mL^−1^)	Yes	1.40 (1.30–1.60)	0.10	2.95 (2.46–3.93)	0.03	1.89 (1.50–4.16)	0.68
	No	1.52 (1.33–1.84)		3.86 (2.82–4.12)		3.95 (1.51–4.14)	
MMP7 (pg·mL^−1^)	Yes	0.51 (0.45–0.63)	0.81	0.65 (0.55–0.82)	0.79	0.65 (0.50–0.86)	0.77
	No	0.52 (0.47–0.63)		0.66 (0.52–0.88)		0.65 (0.50–0.86)	
MMP9 (pg·mL^−1^)	Yes	853.49 (752.04–944.03)	0.44	961.28 (900.34–1082.41)	0.07	993.09 (881.85–1299.96)	0.28
	No	811.19 (713.42–927.06)		1112.75 (931.7–1363.64)		1211.37 (877.75–1362.5)	

IQR: interquartile range; IL: Interleukine; MCP: chemoattractant protein-1; CO: carbon oxide; NO: nitric oxide; MMP: Matrix Metalloproteinases.

**Table 3 cancers-17-01159-t003:** Data from the bronchoalveolar lavage of the operated (oncologic) lung.

Variable	Recurrence	Baseline Median (IQR)	*p*-Value	End of Surgery Median (IQR)	*p*-Value
IL-1 (pg·mL^−1^)	Yes	129.52 (121.10–149.74)	0.510	183.40 (169.34–206.84)	0.606
	No	128.81 (113.68–147.44)		188.72 (167.80–219.65)	
TNFα (pg·mL^−1^)	Yes	15.05 (13.81–15.52)	0.643	23.01 (20.48–24.97)	0.073
	No	14.94 (13.92–16.63)		20.94 (19.98–23.38)	
IL-2 (pg·mL^−1^)	Yes	2.19 (1.99–2.60)	0.321	3.22 (2.96–3.92)	0.315
	No	2.14 (1.95–2.30)		3.16 (2.90–3.79)	
IL-6 (pg·mL^−1^)	Yes	6.33 (5.76–6.67)	0.975	7.60 (7.18–8.06)	0.264
	No	6.25 (5.80–6.92)		7.42 (6.86–7.94)	
IL-10 (pg·mL^−1^)	Yes	40.75 (39.14–42.38)	0.901	42.06 (39.96–44.27)	0.753
	No	40.92 (39.98–42.08)		41.37 (39.67–44.35)	
MCP1 (pg·mL^−1^)	Yes	374.70 (366.26–390.77)	0.750	545.96 (529.67–574.70)	0.232
	No	382.59 (349.05–399.56)		541.09 (517.49–555.15)	
IL-4 (pg·mL^−1^)	Yes	0.41 (0.38–0.43)	0.978	0.86 (0.73–0.93)	0.300
	No	0.41 (0.38–0.43)		0.80 (0.71–0.91)	
IL-7 (pg·mL^−1^)	Yes	3.13 (2.88–3.28)	0.060	5.21 (4.86–5.76)	0.630
	No	3.20 (2.98–3.62)		5.08 (4.96–5.60)	
IL-8 (pg·mL^−1^)	Yes	2.76 (2.51–3.06)	0.436	50.88 (25.79–57.44)	0.503
	No	2.71 (2.05–2.97)		32.55 (26.52–51.49)	
IL-12 (pg·mL^−1^)	Yes	0.07 (0.06–0.07)	0.191	0.13 (0.11–0.15)	0.350
	No	0.07 (0.06–0.08)		0.13 (0.10–0.14)	
VEGF (pg·mL^−1^)	Yes	106.32 (93.31–115.82)	0.150	118.84 (103.85–130.48)	0.085
	No	98.82 (87.22–110.87)		108.63 (98.79–124.53)	
NO (mmHg)	Yes	12.50 (6.15–15.85)	0.394	11.05 (8.01–20.69)	0.358
	No	8.15 (6.02–14.44)		9.49 (7.43–17.41)	
CO (mmHg)	Yes	6.39 (5.71–6.90)	0.767	8.17 (7.82–9.18)	0.506
	No	6.20 (5.77–7.22)		8.30 (7.43–9.10)	
MMP2 (pg·mL^−1^)	Yes	5.03 (4.42–12.58)	0.575	9.77 (8.60–16.50)	0.761
	No	4.89 (4.31–12.00)		9.72 (8.36–15.38)	
MMP3 (pg·mL^−1^)	Yes	3.37 (3.00–3.77)	0.047	9.30 (7.12–9.86)	0.582
	No	3.78 (2.97–4.07)		8.44 (6.96–9.84)	
MMP7 (pg·mL^−1^)	Yes	0.50 (0.47–0.52)	0.368	0.53 (0.51–0.55)	0.755
	No	0.51 (0.47–0.53)		0.53 (0.51–0.56)	
MMP9 (pg·mL^−1^)	Yes	1188.68 (1007.70–1244.50)	0.020	1394.42 (1359.35–1500.73)	0.346
	No	1250.98 (1158.85–1291.16)		1402.43 (1341.95–1433.73)	

IQR: interquartile range; IL: Interleukine; MCP: chemoattractant protein-1; CO: carbon oxide; NO: nitric oxide; MMP: Matrix Metalloproteinases.

**Table 4 cancers-17-01159-t004:** Regression models. Model 1: only clinical variables. Model 2: clinical variables with more bronchoalveolar or blood biomarkers.

Variables	Coefficient (B)	*p*-Value
Model 1		
Type of surgery		0.051
Lobectomy	−2.045	0.015
Segmentectomy	−1.727	0.066
T3/T4 staging	1.739	0.043
Model 2		
Type of surgery		0.021
Lobectomy	−4.16	0.006
Segmentectomy	−3.493	0.028
IL6 plasma at 6 h	−1.718	0.02
MCP1 plasma at 18 h	0.021	0.032
IL6/IL10 plasma at 6 h	0.111	0.014
MMP3 plasma at baseline	2.93	0.023
TNFα BAL at surgery end	0.428	0.016
MMP9 BAL at surgery start	−0.006	0.017
MMP9 BAL at surgery end	0.009	0.008

## Data Availability

The data generated in the present study may be requested from the corresponding author.

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
