# Peer review of "Exploring the Relationship Between Perioperative Inflammatory Biomarkers and Oncological Recurrence in Patients Undergoing Pulmonary Cancer Surgery"

_cancers, 2025, doi:10.3390/cancers17071159_

Round 1
Reviewer 1 Report
Comments and Suggestions for Authors
This work reveals the association between perioperative expression of inflammatory biomarkers and lung cancer recurrence. The study presents data accumulated over 10 years. The research is interesting and well-presented. However, a major flaw lies in the Materials and Methods section, which requires detailed experimental procedures.
There are numerous typographical and language corrections needed.
Materials and Methods:
A detailed experimental procedure is required for sample collection, storage, preparation, and cytokine examination.
Specific Revisions:
- Line 3: Revise “Bi-omarkers” to “Bio-markers.”
- Line 5: All authors' affiliations must be indicated with superscripted numerals after their names. Additionally, the corresponding authors must be marked with an asterisk.
- Abstract: The subsection Discussion must be replaced with Conclusions.
- Line 27: Use either peri-operative or perioperative consistently throughout the manuscript.
- Line 29: Remove the repeated phrase "we conducted a retrospective cohort study…"
- Line 33: Use either pro-inflammatory or proinflammatory consistently.
- Line 36: Use "identified" instead of the current wording.
- Line 125: Revise "cytokine blood level" for better clarity.
- Line 143: Use either "h" or "hours" consistently; MDPI journals prefer "h."
- Line 163: The phrase "Table…" is unnecessary and should be removed.
- Line 201: Remove the word "see."
- Line 330: Revise references according to MDPI format.
Many sentences are confusing, and there are numerous typographical and grammatical errors. The authors are advised to have the manuscript reviewed by a native speaker.
Author Response
Specific Revisions:
- Line 3: Revise “Bi-omarkers” to “Bio-markers.”
RESPONSE (R) Yes: we´ve revised
- Line 5: All authors' affiliations must be indicated with superscripted numerals after their names. Additionally, the corresponding authors must be marked with an asterisk.
R: Yes, we´ve modified
- Abstract: The subsection Discussion must be replaced with Conclusions.
R: yes, now we´ve corrected
- Line 27: Use either peri-operative or perioperative consistently throughout the manuscript.
R: Yes, we´ll use consistently perioperative (no peri-operative)
- Line 29: Remove the repeated phrase "we conducted a retrospective cohort study…"
R: Yes we´ve removed
- Line 33: Use either pro-inflammatory or proinflammatory consistently.
R: Yes, now we use proinflammatory consistently
- Line 36: Use "identified" instead of the current wording.
R: Yes we´ve changed
- Line 125: Revise "cytokine blood level" for better clarity.
R: Yes. We´ve changed
- Line 143: Use either "h" or "hours" consistently; MDPI journals prefer "h."
R: Yes, now we use “h”
- Line 163: The phrase "Table…" is unnecessary and should be removed.
R: Yes, now we´ve removed
- Line 201: Remove the word "see."
R: Yes, we´ve removed
- Line 330: Revise references according to MDPI format.
R: Yes, now we´ve corrected acording MDPI format
Comments on the Quality of English Language: Many sentences are confusing, and there are numerous typographical and grammatical errors. The authors are advised to have the manuscript reviewed by a native speaker.
Response: Now, quality of manuscript English language has been reviewed
Reviewer 2 Report
Comments and Suggestions for Authors
The paper investigates the role of perioperative inflammatory biomarkers in predicting lung cancer recurrence after surgical treatment. The authors conducted a retrospective study involving 93 lung cancer patients followed over ten years, measuring inflammatory biomarkers in blood and bronchoalveolar lavage (BAL) samples before and after surgery. They identified that certain cytokines, metalloproteinases (particularly MMP-2, MMP-3, MMP-9), and their ratios (IL-6/IL-10) significantly improved the prediction accuracy of recurrence when combined with clinical factors (type of surgery and tumor stage). The authors concluded that monitoring perioperative inflammatory markers can significantly enhance recurrence prediction, offering opportunities for early intervention and personalized patient management.
Figure 1: ROC Curves for Logistic Regression Models
Authors should clarify the labels and legends to explicitly indicate what each curve represents (for example, "Model 1: Clinical only," "Model 2: Clinical + biomarkers"). Additionally, they should include a numeric summary (AUC values) directly on the figure for improved readability. Adding confidence intervals to the ROC curves will also enhance statistical interpretation.
Suggestions for additional figures
New Figure 2 (Suggested): Blood Cytokine Levels.
Visualize the dynamics of significant cytokines (e.g., IL-6, MCP-1) at different time points (baseline, 6h, and 18h post-surgery), comparing the recurrence and non-recurrence groups.
New Figure 3 (Suggested): Cytokine Levels in BAL Fluid.
Display the levels of cytokines (TNFα, MMP-9, MMP-3) at baseline and immediately after surgery in patients with recurrence compared to those without recurrence.
Author Response
Figure 1: ROC Curves for Logistic Regression Models
Authors should clarify the labels and legends to explicitly indicate what each curve represents (for example, "Model 1: Clinical only," "Model 2: Clinical + biomarkers"). Additionally, they should include a numeric summary (AUC values) directly on the figure for improved readability. Adding confidence intervals to the ROC curves will also enhance statistical interpretation.
R: Yes, we´ve made a new figure (number 3)
Suggestions for additional figures
New Figure 2 (Suggested): Blood Cytokine Levels.
Visualize the dynamics of significant cytokines (e.g., IL-6, MCP-1) at different time points (baseline, 6h, and 18h post-surgery), comparing the recurrence and non-recurrence groups.
R: Now, we´ve made a new figure (number 1)
New Figure 3 (Suggested): Cytokine Levels in BAL Fluid.
Display the levels of cytokines (TNFα, MMP-9, MMP-3) at baseline and immediately after surgery in patients with recurrence compared to those without recurrence
R: Now, we´ve made a new figure (number 2)
Round 2
Reviewer 1 Report
Comments and Suggestions for Authors
The authors have revised the manuscript according to my comments.The manuscript now has the merit for publication in Cancers.